# Gold(I) and Silver(I) Complexes Containing Hybrid Sulfonamide/Thiourea Ligands as Potential Leishmanicidal Agents

**DOI:** 10.3390/pharmaceutics16040452

**Published:** 2024-03-25

**Authors:** Alice P. Borges, Malu M. S. Obata, Silvia H. Libardi, Rafael O. Trevisan, Victor M. Deflon, Ulrich Abram, Francis B. Ferreira, Luiz Antônio S. Costa, Antonio O. T. Patrocínio, Marcos V. da Silva, Júlio C. Borges, Pedro I. S. Maia

**Affiliations:** 1Bioactive Compounds Development Research Group, Federal University of Triângulo Mineiro, Av. Dr. Randolfo Borges 1400, Uberaba 38025-440, MG, Brazil; aliceborges98@hotmail.com; 2Department of Microbiology, Immunology and Parasitology, Federal University of Triângulo Mineiro, Av. Getúlio Guarita, Uberaba 38025-440, MG, Brazil; malu.mateus_21@hotmail.com (M.M.S.O.); rafaelotrevisan@gmail.com (R.O.T.); marcos.silva@uftm.edu.br (M.V.d.S.); 3São Carlos Institute of Chemistry, University of São Paulo, Av. Trabalhador São Carlense, 400, São Carlos 13566-590, SP, Brazil; libardish@gmail.com (S.H.L.); deflon@iqsc.usp.br (V.M.D.); borgesjc@iqsc.usp.br (J.C.B.); 4Institute of Chemistry and Biochemistry, Freie Universität Berlin, Fabeckstr. 34/36, 14195 Berlin, Germany; ulrich.abram@fu-berlin.de; 5Associated Faculties of Uberaba, Av. do Tutuna 720, Uberaba 38061-500, MG, Brazil; francisbferreira@gmail.com; 6Computational Chemistry Research Group, Institute of Exact Sciences, Federal University of Juiz de Fora, Campus Universitário s/n Martelos, Juiz de Fora 36036-900, MG, Brazil; luiz.costa@ufjf.br; 7Laboratory of Photochemistry and Material Science, Institute of Chemistry, Federal University of Uberlândia, Av. João Naves de Ávila 2121, Uberlândia 38400-902, MG, Brazil; otaviopatrocinio@ufu.br

**Keywords:** hybrid compounds, thioureas, sulfonamides, leishmanicidal activity, molecular docking, enzymatic studies

## Abstract

Leishmaniasis is a group of parasitic diseases with the potential to infect more than 1 billion people; however, its treatment is still old and inadequate. In order to contribute to changing this view, this work consisted of the development of complexes derived from M^I^ metal ions with thioureas, aiming to obtain potential leishmanicidal agents. The thiourea ligands (HL^R^) were obtained by reactions of *p*-toluenesulfohydrazide with R-isothiocyanates and were used in complexation reactions with Ag^I^ and Au^I^, leading to the formation of complexes of composition [M(HL^R^)_2_]X (M = Ag or Au; X = NO_3_^−^ or Cl^−^). All compounds were characterized by FTIR, ^1^H NMR, UV-vis, emission spectroscopy and elemental analysis. Some representatives were additionally studied by ESI-MS and single-crystal XRD. Their properties were further analyzed by DFT calculations. Their cytotoxicity on Vero cells and the extracellular leishmanicidal activity on *Leishmania infantum* and *Leishmania braziliensis* cells were evaluated. Additionally, the interaction of the complexes with the Old Yellow enzyme of the *L. braziliensis* (LbOYE) was examined. The biological tests showed that some compounds present remarkable leishmanicidal activity, even higher than that of the standard drug Glucantime, with different selectivity for the two species of *Leishmania*. Finally, the interaction studies with LbOYE revealed that this enzyme could be one of their biological targets.

## 1. Introduction

Leishmaniasis is considered one of the most neglected tropical and subtropical parasitic diseases and a problem of public health in almost 90 countries [1,2,3,4]. An increasing number of cases per year have been pointed out (700 thousand to 1 million), which is related to the migratory flow of refugees from regions of conflicts to tropical and subtropical countries [1,2]. The transmission of Leishmaniasis occurs mainly in endemic areas, where the protozoa are spread when female phlebotomine sandflies bite infected animals and then humans [5,6]. Among the hosts of these parasites, domestic animals, mostly dogs, are the main source of infection for people who are in close contact with them [7,8,9]. The most common prescriptions for the treatment of Leishmaniasis are based on pentavalent antimonials (Sb^V^), like Glucantime and Pentostam, which have in their composition meglumine antimoniate and sodium stibogluconate, respectively [2,10,11]. However, this is not a good alternative since, although offering a relatively low toxicity, these drugs are not effective in many cases; in addition, the growing incidence of resistance limits their use in endemic areas [10,11]. Therapy with the second-line drugs paromomycin, amphotericin B and miltefosine presents even more problems, which comprises the need of parenteral applications and severe side effects, including teratogenic potential in the case of miltefosine [10]. Although a lot of effort has been made, so far there is no effective vaccine in clinical trials [12]. Therefore, new therapeutic alternatives for treating leishmaniasis are truly needed. In an effort to change this situation, several metal-based compounds, including silver and gold complexes (Figure 1B), have been studied and identified as antileishmanial drug prototypes and some of them have shown in vivo activity [13,14,15]. Despite these recent advances in antileishmanial drug discovery, to our knowledge, no metallodrug has passed clinical trials.

In order to have a good metallodrug candidate, the choice of the ligand is very important since the organic molecule is responsible for stabilizing the metal center, avoiding losing the compounds for non-target biomolecules, which in many cases are also responsible for the activity. Thioureas and sulfonamides are two classes of compounds with a broad spectrum of pharmacological applications, among which the following stand out: antibacterial, antitumor, antiparasitic, antiviral and anticonvulsant [16,17,18]. The union of these classes of compounds in a single molecule by a covalent bond allows one, therefore, to obtain hybrid molecules. In addition to enhancing activity, these hybrid molecules can lead to a synergistic effect between two classes in the same compound and to a mechanism of action on more than one target. Our research uses, as a strategy, the development of hybrid molecules by the combination of bioactive groups such as sulfonamides, thioureas, oximes, carbazates, thiosemicarbazides and other classes of molecules with known biological activity [19,20,21,22]. Additionally, in order to achieve multiple mechanisms of action against the parasite, the coordination of these bioactive ligands to metal ions has been accomplished, producing compounds with enhanced biological activity when compared to the free ligands [23]. This hypothesis was confirmed in a previous work, where Cu^I^ complexes derived from the HL^R^ ligands (Figure 1C,D) presented better trypanocidal activity against the amastigote form of the *Tulahuen LacZ* strain of *Trypanosoma cruzi*, the causative agent of Chagas disease [24]. Notably, the cytotoxicity of the compounds on LLC-MK2 cells did not increase upon complexation. In addition to our work, only a few complexes containing such type of ligands have been reported up to now, which were not structurally characterized [25,26].

The promising findings found for silver and gold complexes during the last decade, together with our previous encouraging results for the HL^R^ derivatives, gave us the interest in seeking to verify the influence of changing the peripheral structure of organic molecules and also the metal center on antiparasitic activity. In this context, herein, new ligands were prepared, coordinated to Ag^I^ and Au^I^ ions and evaluated against leishmania strains. Many mechanisms of action have been proposed for transition metal complexes [27,28]; however, the trypanothione reductase system is the most studied for such type of compounds [15,29,30,31]. Herein, we decided to investigate the LbOYE as a molecular target of the new coordination compounds, including both experimental and theoretical methods.

## 2. Materials and Methods

### 2.1. Materials

All solvents were obtained commercially and used without prior treatment. The reagents, *p*-toluenesulfohydrazide, cyclohexyl isothiocyanate, phenyl isothiocyanate, 4-fluorophenyl isothiocyanate, 4-chlorophenyl isothiocyanate, 4-nitrophenyl isothiocyanate, allyl isothiocyanate, AgNO_3_ and [AuCl(DMS)] were obtained commercially (Sigma Aldrich, Cotia, Brazil). HL1^Ch^, HL2^Ph^, [Ag(HL1^Ch^)_2_]NO_3_ (**Ag1**) and [Ag(HL2^Ph^)_2_]NO_3_ (**Ag2**) were obtained as previously reported [22].

### 2.2. General Methods

The melting points were determined using a PF1500 FARMA-GEHAKA instrument (São Paulo, Brazil). The elemental analyses (CHNS) were determined using Heraeus vario EL equipment (Langenselbold, Germany). The conductivities of the complexes were measured with an CG1800-GEHAKA conductometer (São Paulo, Brazil). The vibrational spectra were carried out using a PerkinElmer spectrophotometer model FT-IR Frontier Single Range (Waltham, MA, USA) in the region between 4000 and 200 cm^−1^. The sample analyzes were performed in the solid state using the Attenuated Total Reflectance (ATR) accessory with diamond crystal. The NMR spectra were recorded on a Bruker Ascend 400 (Ettlingen, Germany) or on a JEOL 400 MHz multinuclear spectrometer (Tokyo, Japan), operating at 400 and 377 MHz for ^1^H and ^19^F{1H}, respectively. The electronic spectra were measured with a Shimadzu UV-2501 spectrophotometer (Tokyo, Japan) in methanol or acetonitrile solutions. The emission spectra were obtained in a Horiba Fluoromax-4 fluorimeter (Kyoto, Japan) at room temperature. The excitation and emission slits were set to 6 nm. Solutions were purged with argon for five minutes before the measurements. The measurements at 77 K glass medium were performed in a Dewar flask with liquid nitrogen in a solvent mixture of 4:1 ethanol/methanol (*v*/*v*). The positive-mode electrospray ionization mass spectra (ESI-MS) were recorded with an Agilent 6210 ESI-TOF (Agilent Technologies Santa Clara, CA, USA). All MS results are given in the form: *m*/*z*, assignment.

### 2.3. Preparation of the Compounds

#### 2.3.1. Synthesis of HL^R^ Ligands

The HL^R^ (R = F, Cl, NO_2_ and allyl) ligands were formed by a method adopted from procedures in the literature [32,33,34]. Briefly, 3.80 mmol of the desired R-isothiocyanate were added to a solution containing 1 equivalent of *p*-toluenesulfohydrazide 708 mg (3.80 mmol) dissolved in 10 mL of ethanol. The reaction mixtures were left under stirring at 60 °C for 3 h until the formation of colorless precipitates was observed. After keeping the reaction mixtures in the freezer for 3 days, the solids were filtered, washed with *n*-hexane and dried under vacuum. The solids were recrystallized from a dichloromethane:ethanol mixture in a 2:1 (*v*/*v*) ratio. Crystals were obtained after slow evaporation of the resulting solutions for all compounds.

*N-(4-fluorophenyl)-2-tosylhydrazinocarbothioamide* (HL3^FPh^)–Color: Colorless. Yield: 85% (1.09 g). M.P.: 185–186 °C. Elemental Analysis calculated for C_14_H_14_FN_3_O_2_S_2_ (339.41 g mol^−1^): C: 49.54; H: 4.16; N: 12.38; S: 18.89%. Found: C: 49.58; H: 4.17; N: 12.43; S: 18.57%. IR (ATR/cm^–1^): 3320, 3240, 3133 ν(N-H), 1596 ν(C=N), 1543 ν(C=C), 1333 ν(S=O), 1156 ν(S=O) and 814 ν(C=S). ^1^H NMR (400 MHz, DMSO-*d_6_*, δ/ppm): 2.40 (s, 3H, CH_3_), 7.10–7.14 (m, 2H, CH-FPh), 7.28–7.31 (m, 2H, CH-FPh), 7.42 (d, ^3^*J* = 8 Hz, 2H, CH-*p*-toluene), 7.76 (d, ^3^*J* = 8 Hz, 2H, CH-*p*-toluene), 9.66 (s, 1H, NH), 9.75 (s, 1H, NH) and 9.94 (s, 1H, NH). ^19^F NMR (377 MHz, DMSO-*d_6_*, δ/ppm): −118.43 (s). UV-vis, solution of CH_3_OH concentration 10^−5^ mol L^−1^ [λ_max_/ε (L mol^−1^ cm^−1^)]: 227 nm (69,185) and 265 nm (29,711).

*N-(4-chlorophenyl)-2-tosylhydrazinocarbothioamide* (HL4^ClPh^)–Color: Colorless. Yield: 87% (1.17 g). M.P.: 183–184 °C. Elemental Analysis calculated for C_14_H_14_ClN_3_O_2_S_2_ (355.86 g mol^−1^): C: 47.25; H: 3.97; N: 11.81; S: 18.02%. Found: C: 47.24; H: 3.96; N: 11.91; S: 17.71%. IR (ATR/cm^–1^): 3306, 3290, 3145 ν(N-H), 1600, 1584 ν(C=N), 1532, 1510 ν(C=C), 1341 ν(S=O), 1161 ν(S=O) and 806 ν(C=S). ^1^H NMR (400 MHz, DMSO-*d_6_*, δ/ppm): 2.39 (s, 3H, CH_3_), 7.31–7.39 (m, 4H, CH-ClPh), 7.41 (d, ^3^*J* = 8 Hz, 2H, CH-*p*-toluene), 7.75 (d, ^3^*J* = 8 Hz, 2H, CH-*p*-toluene), 9.72 (s, 1H, NH), 9.82 (s, 1H, NH) and 9.96 (s, 1H, NH). UV-vis, solution of CH_3_OH concentration 10^−5^ mol L^−1^ [λ_max_/ε (L mol^−1^ cm^−1^)]: 228 nm (65,744) and 273 nm (26,616).

*N-(4-nitrophenyl)-2-tosylhydrazinocarbothioamide* (HL5^NO2Ph^)–Color: Colorless. Yield: 80% (1.11 g). M.P.: 181–183 °C. Elemental Analysis calculated for C_14_H_14_N_4_O_4_S_2_ (366.41 g mol^−1^): C: 45.89; H: 3.85; N: 15.29; S: 17.50 %. Found: C: 45.89; H: 3.87; N: 15.64; S: 17.63%. IR (ATR/cm^–1^): 3308, 3240, 3136 ν(N-H), 1615, 1597 ν(C=N), 1548 ν(C=C), 1424 ν(N-O), 1331 ν(S=O), 1158 ν(S=O) and 807 ν(C=S). ^1^H NMR (400 MHz, DMSO-*d_6_*, δ/ppm): 2.35 (s, 3H, CH_3_), 7.39 (d, ^3^*J* = 8 Hz, 2H, CH-*p*-toluene), 7.75 (d, ^3^*J* = 8 Hz, 2H, CH-*p*-toluene), 7.80 (d, ^3^*J* = 8 Hz, 2H, CH-NO_2_Ph), 8.17 (d, ^3^*J* = 12 Hz, 2H, CH-NO_2_Ph), 10.06 (s, 1H, NH), 10.10 (s, 1H, NH) and 10.17 (s, 1H, NH). UV-vis, solution of CH_3_OH concentration 10^−5^ mol L^−1^ [λ_max_/ε (L mol^−1^ cm^−1^)]: 227 nm (46,030), 270 nm (19,579) and 331 nm (13,099).

*N-allyl-2-tosylhydrazinocarbothioamide* (HL6^Al^)–Color: Colorless. Yield: 69 % (0.74 g). M.P.: 166–167 °C. Elemental Analysis calculated for C_11_H_15_N_3_O_2_S_2_ (285.39 g mol^−1^): C: 46.29; H: 5.30; N: 14.72; S: 22.47 %. Found: C: 46.15; H: 5.35; N: 14.85; S: 22.51 %. IR (ATR/cm^–1^): 3358, 3160 ν(N-H), 1600 ν(C=N), 1557 ν(C=C), 1348 ν(S=O), 1170 ν(S=O) and 809 ν(C=S). ^1^H NMR (400 MHz, DMSO-*d_6_*, δ/ppm): 2.41 (s, 3H, CH_3_), 4.06 (t, 2H, CH_2_-NH), 5.00–5.11 (m, 2H, CH_2_-CH), 5.69–5.79 (m, 1H, CH-CH_2_), 7.41 (d, ^3^*J* = 8 Hz, 2H, CH-*p*-toluene), 7.70 (d, ^3^*J* = 8 Hz, 2H, CH-*p*-toluene), 8.18 (t, 1H, NH-CH_2_), 9.35 (s, 1H, NH) and 9.82 (s, 1H, NH). UV-vis, solution of CH_3_OH concentration 10^−5^ mol L^−1^ [λ_max_/ε (L mol^−1^ cm^−1^)]: 228 nm (53,510) and 246 nm (41,523).

#### 2.3.2. Synthesis of [Ag(HL^R^)_2_]NO_3_ Complexes

The desired ligand (0.20 mmol) was added to a solution containing the precursor AgNO_3_ (0.10 mmol, 17 mg) dissolved in ethanol (4 mL). The solution was maintained under stirring for 24 h, forming colorless precipitates. The solids were filtered, washed with *n*-hexane and dried under vacuum.

[Ag(HL3^FPh^)_2_]NO_3_ (**Ag3**): Color: Colorless. Yield: 59% (50 mg). M.P.: 159–160 °C. Elemental Analysis calculated for C_28_H_28_AgF_2_N_7_O_7_S_4_ (848.69 g mol^−1^): C: 39.63; H: 3.33; N: 11.55; S: 15.11%. Found: C: 39.73; H: 3.34; N: 11.58; S: 15.19%. IR (ATR/cm^–1^): 3233, 3183, 3156 ν(N-H), 1609 ν(C=N), 1559 ν(C=C), 1416 ν(N-O), 1332 ν(S=O), 1160 ν(S=O) and 786 ν(C=S). ^1^H NMR (DMSO-*d_6_*, δ/ppm): 2.38 (s, 6H, CH_3_), 7.13 (d, *J* = 8 Hz, 4H, CH-*p*-toluene), 7.22–7.41 (m, 12H, CH-FPh), 7.73 (d, *J* = 8 Hz, 4H, CH-*p*-toluene), 10.29 (s, 2H, NH), 10.35 (s, 2H, NH) and 10.44 (s, 2H, NH). ^19^F NMR (377 MHz, DMSO-*d_6_*, δ/ppm): −115.15 (s). ESI/MS (*m*/*z*): 785.0040, [M]^+^ (calcd. 785.0074). UV-vis, solution of CH_3_CN concentration 10^−6^ mol L^−1^ [λ_max_/ε (L mol^−1^ cm^−1^)]: 228 nm (22,532) and 265 nm (9229). Molar Conductivity: (10^−3^ M, DMSO): 4.48 μS cm^−1^ mol^−1^.

[Ag(HL4^ClPh^)_2_]NO_3_ (**Ag4**): Color: Colorless. Yield: 76% (67 mg). M.P.: 162–164 °C. Elemental Analysis calculated for C_28_H_28_AgCl_2_N_7_O_7_S_4_ (881.60 g mol^−1^): C: 38.15; H: 3.20; N: 11.12; S: 14.55%. Found: C: 40.67; H: 3.30; N: 10.54; S: 15.75%. IR (ATR/cm^–1^): 3228, 3184, 3166 ν(N-H), 1593 ν(C=N), 1550 ν(C=C), 1409 ν(N-O), 1332 ν(S=O), 1160 ν(S=O) and 784 ν(C=S). ^1^H NMR (DMSO-*d_6_*, δ/ppm): 2.37 (s, 6H, CH_3_), 7.04–7.47 (m, 12H, CH-ClPh + CH-*p*-toluene), 7.76 (d, *J* = 8 Hz, 4H, CH-*p*-toluene), 8.32 (s, 1H, NH), 8.93 (s, 1H, NH) and 10.14 (s, 4H, NH). UV-vis, solution of CH_3_CN concentration 10^−6^ mol L^−1^ [λ_max_/ε (L mol^−1^ cm^−1^)]: 228 nm (26,078) and 278 nm (9379). Molar Conductivity: (10^−3^ M, DMSO): 4.25 μS cm^−1^ mol^−1^.

[Ag(HL5^NO2Ph^)_2_]NO_3_ (**Ag5**): Color: Colorless. Yield: 79% (71 mg). M.P.: 158–160 °C. Elemental Analysis calculated for C_28_H_28_AgN_9_O_11_S_4_ (902.70 g mol^−1^): C: 37.25; H: 3.13; N: 13.96; S: 14.21%. Found: C: 36.95; H: 3.00; N: 13.62; S: 13.84%. IR (ATR/cm^–1^): 3385, 3306, 3217, 3155 ν(N-H), 1619, 1596 ν(C=N), 1570 ν(C=C), 1418, 1400 ν(N-O), 1318 ν(S=O), 1154 ν(S=O) and 760 ν(C=S). ^1^H NMR (DMSO-*d_6_*, δ/ppm): 2.37 (s, 6H, CH_3_), 7.42 (d, *J* = 8 Hz, 4H, CH-*p*-toluene), 7.51 (d, *J* = 8 Hz, 4H, CH-*p*-toluene), 7.71–8.30 (m, 8H, CH-NO_2_Ph), 9.16 (s, 1H, NH), 9.30 (s, 1H, NH) and 10.26–10.82 (m, 4H, NH). UV-vis, solution of CH_3_CN concentration 10^−6^ mol L^−1^ [λ_max_/ε (L mol^−1^ cm^−1^)]: 227 nm (37,436), 280 nm (16,203) and 332 nm (15,907). Molar Conductivity: (10^−3^ M, DMSO): 20.4 μS cm^−1^ mol^−1^.

[Ag(HL6^Al^)_2_]NO_3_ (**Ag6**): Color: Colorless. Yield: 68% (50 mg). M.P.: 183–187 °C. Elementary Analysis calculated for C_22_H_30_AgN_7_O_7_S_4_ (740.64 g mol^−1^): C: 35.68; H: 4.08; N: 13.24; S: 17.32%. Found: C: 35.92; H: 4.21; N: 13.26; S: 17.33 %. IR (ATR/cm^–1^): 3265, 3225 ν(N-H), 1646 ν(C=N), 1568 ν(C=C), 1418 ν(N-O), 1327 ν(S=O), 1134 ν(S=O) and 781 ν(C=S). ^1^H NMR (DMSO-*d_6_*, δ/ppm): 2.38 (s, 6H, CH_3_), 4.05 (m, 4H, CH_2_-NH), 4.94–5.22 (m, 4H, CH_2_-CH), 5.61–5.90 (m, 2H, CH-CH_2_), 7.37 (d, *J* = 8 Hz, 4H, CH-*p*-toluene), 7.68 (d, *J* = 8 Hz, 4H, CH-*p*-toluene), 8.31 (s, 2H, NH) and 9.48 (s, 2H, NH). UV-vis, solution of CH_3_CN concentration 10^−6^ mol L^−1^ [λ_max_/ε (L mol^−1^ cm^−1^)]: 228 nm (72,213) and 251 nm (56,659). Molar Conductivity: (10^−3^ M, DMSO): 6.60 μS cm^−1^ mol^−1^.

#### 2.3.3. Synthesis of [Au(HL^R^)_2_]Cl Complexes

A total of 0.20 mmol of the desired ligand was added to a solution containing 0.10 mmol (30 mg) of the precursor [AuCl(DMS)] dissolved in acetonitrile (4 mL). The reactions were maintained under stirring for 24 h, leading to the formation of colorless precipitates which were filtered, washed with *n*-hexane and dried under vacuum. Then, the solids were recrystallized in a dichloromethane/acetonitrile mixture in a 2:1 (*v*/*v*) ratio. Colorless crystals were obtained by slow evaporation of the solutions at room temperature in the absence of light.

[Au(HL1^Ch^)_2_]Cl (**Au1**): Color: Colorless. Yield: 73% (65 mg). M.P.: 183–187 °C. Elemental Analysis calculated for C_28_H_42_AuClN_6_O_4_S_4_ (887.35 g mol^−1^): C: 37.90; H: 4.77; N: 9.47; S: 14.45%. Found: C: 37.90; H: 4.86; N: 9.48; S: 14.61%. IR (ATR/cm^–1^): 3307, 3286, 3193 ν(N-H), 1600 ν(C=N), 1575 ν(C=C), 1330 ν(S=O), 1164 ν(S=O) and 789 ν(C=S). ^1^H NMR (DMSO-*d_6_*, δ/ppm): 1.09–1.68 (m, 20H, CH_2_), 2.40 (s, 6H, CH_3_), 3.91–3.99 (m, 2H, CH-CH_2_), 7.43 (d, *J* = 8 Hz, 4H, CH-*p*-toluene), 7.73 (d, *J* = 8 Hz, 4H, CH-*p*-toluene), 8.26 (s, 1H, H) and 9.96 (s, 3H, NH). ESI^+^ MS (*m*/*z*): 851.1810, [M]^+^ (calcd. 851.1816); 1374.2460 [M + AuL1^Ch^]^+^ (calcd. 1374.2479). UV-vis, solution of MeOH concentration 10^−6^ mol L^−1^ [λ_max_/ε (L mol^−1^ cm^−1^)]: 227 nm (78,234) and 255 nm (40,522). Molar Conductivity: (10^−3^ M, CH_3_OH): 65.1 μS cm^−1^ mol^−1^.

[Au(HL2^Ph^)_2_]Cl (**Au2**): Color: Colorless. Yield: 89% (78 mg). M.P.: 171–173 °C. Analysis calculated for C_28_H_30_AuClN_6_O_4_S_4_ (875.25 g mol^−1^): C: 38.42; H: 3.45; N: 9.60; S: 14.65 %. Found: C: 38.69; H: 3.55; N: 9.64; S: 14.68%. IR (ATR/cm^–1^): 3233, 3161, 3131 ν(N-H), 1597 ν(C=N), 1563 ν(C=C), 1352 ν(S=O), 1162 ν(S=O) and 798 ν(C=S). ^1^H NMR (DMSO-*d_6_*, δ/ppm): 2.40 (s, 6H, CH_3_), 7.12–7.40 (m, 10H, CH-phenyl), 7.43 (d, ^3^*J* = 8 Hz, 4H, CH-*p*-toluene), 7.78 (d, ^3^*J* = 8 Hz, 4H, CH-*p*-toluene) and 10.23 (s_a_, 5H, NH). ESI^+^ MS (*m*/*z*): 839.0767, [M]^+^ (calcd. 839.0877); 1356.0880, [M + AuL2^Ph^]^+^ (calcd. 1356.1070). UV-vis, solution of MeOH concentration 10^−6^ mol L^−1^ [λ_max_/ε (L mol^−1^ cm^−1^)]: 227 nm (91,320) and 270 nm (53,975). Molar Conductivity: (10^−3^ M, CH_3_OH): 74.9 μS cm^−1^ mol^−1^.

[Au(HL3^FPh^)_2_]Cl (**Au3**): Color: Colorless. Yield: 81% (74 mg). M.P.: 192–193 °C. Analysis calculated for C_28_H_28_AuClF_2_N_6_O_4_S_4_ (911.24 g mol^−1^): C: 36.91; H: 3.10; N: 9.22; S: 14.08%. Found: C: 36.98; H: 3.18; N: 9.24; S: 14.07%. IR (ATR/cm^–1^): 3235, 3202, 3160 ν(N-H), 1613, 1598 ν(C=N), 1571 ν(C=C), 1324 ν(S=O), 1156 ν(S=O) and 773 ν(C=S). ^1^H NMR (DMSO-*d_6_*, δ/ppm): 2.41 (s, 6H, CH_3_), 7.10–7.28 (m, 8H, CH-FPh), 7.44 (d, ^3^*J* = 8 Hz, 4H, CH-*p*-toluene), 7.78 (d, ^3^*J* = 8 Hz, 4H, CH-*p*-toluene) and 10.26 (s_a_, 6H, NH). ESI/MS (*m*/*z*): 875.0760, [M]^+^ (calcd. 875.0688); 1410.0908, [M + AuL3^FPh^]^+^ (calcd 1410.0788). UV-vis, solution of MeOH concentration 10^−6^ mol L^−1^ [λ_max_/ε (L mol^−1^ cm^−1^)]: 227 nm (32,697) and 264 nm (14,506). Molar Conductivity: (10^−3^ M, CH_3_OH): 84.3 μS cm^−1^ mol^−1^.

[Au(HL4^ClPh^)_2_]Cl (**Au4**): Color: Colorless. Yield: 87% (82 mg). M.P.: 194–195 °C. Elemental analysis calculated for C_28_H_28_AuCl_3_N_6_O_4_S_4_ (944.14 g mol^−1^): C: 35.62; H: 2.99; N: 8.90; S: 13.58%. Found: C: 35.67; H: 2.99; N: 8.95; S: 13.58%. IR (ATR/cm^–1^): 3222, 3158, 3119 ν(N-H), 1591 ν(C=N), 1552 ν(C=C), 1351 ν(S=O), 1160 ν(S=O) and 798 ν(C=S). ^1^H NMR (DMSO-*d_6_*, δ/ppm): 2.40 (s, 6H, CH_3_), 7.13–7.51 (m, 12H, CH-ClPh + CH-*p*-toluene), 7.77 (d, 2H, ^3^*J* = 8 Hz, CH-*p*-toluene) and 10.26 (s_a_, 6H, NH). ESI^+^ MS (*m*/*z*): 907.0080, [M]^+^ (calcd. 907.0097). UV-vis, solution of MeOH concentration 10^−6^ mol L^−1^ [λ_max_/ε (L mol^−1^ cm^−1^)]: 225 nm (33,535) and 270 nm (13,702). Molar Conductivity: (10^−3^ M, DMSO): 5.98 μS cm^−1^ mol^−1^.

[Au(HL5^NO2Ph^)_2_]Cl (**Au5**): Color: Colorless. Yield: 86% (83 mg). M.P.: 192–193 °C. Elemental analysis calculated for C_28_H_28_AuClN_8_O_8_S_4_ (965.25 g mol^−1^): C: 34.84; H: 2.92; N: 11.61; S: 13.29%. Found: C: 34.52; H: 2.89; N: 11.80; S: 13.54%. IR (ATR/cm^–1^): 3222, 3194, 3156 ν(N-H), 1618, 1596 ν(C=N), 1570 ν(C=C), 1426 ν(N-O), 1341 ν(S=O), 1158 ν(S=O) and 773 ν(C=S). ^1^H NMR (DMSO-*d_6_*, δ/ppm): 2.37 (s, 6H, CH_3_), 7.25–7.73 (m, 8H, CH-NO_2_Ph + CH-*p*-toluene), 7.77 (d, ^3^*J* = 8 Hz, 4H, CH-*p*-toluene), 7.93–8.31 (m, 4H, CH-phenyl) and 10.26 (s_a_, 4H, NH). UV-vis, solution of MeOH concentration 10^−6^ mol L^−1^ [λ_max_/ε (L mol^−1^ cm^−1^)]: 223 nm (29,670), 268 (17,210) and 364 nm (7623). Molar Conductivity: (10^−3^ M, DMSO): 30.4 μS cm^−1^ mol^−1^.

[Au(HL6^Al^)_2_]Cl (**Au6**): Color: Colorless. Yield: 83% (67 mg). M.P.: 197–198 °C. Elemental analysis calculated for C_22_H_30_AuClN_6_O_4_S_4_ (803.19 g mol^−1^): C: 32.90; H: 3.76; N: 10.46; S: 15.97%. Found: C: 32.97; H: 3.79; N: 10.46; S: 15.97%. IR (ATR/cm^–1^): 3261, 3192 ν(N-H), 1641 ν(C=N), 1584 ν(C=C), 1350 ν(S=O), 1160 ν(S=O) and 795 ν(C=S). ^1^H NMR (DMSO-*d_6_*, δ/ppm): 2.41 (s, 6H, CH_3_), 4.09 (t, 4H, CH_2_-NH), 5.06–5.16 (m, 4H, CH_2_-CH), 5.70–5.82 (m, 2H, CH-CH_2_), 7.43 (d, *J* = 8 Hz, 4H, CH-*p*-toluene), 7.73 (d, *J* = 8 Hz, 4H, CH-*p*-toluene), 8.98 (s_a_, 2H, NH) and 10.06 (s_a_, 4H, NH). ESI^+^ MS (*m*/*z*): 767.0784, [M]^+^ (calcd. 767.0877; 1248.0891, [M + AuL6^Al^]^+^ (calcd. 1248.1070). UV-vis, solution of MeOH concentration 10^−6^ mol L^−1^ [λ_max_/ε (L mol^−1^ cm^−1^)]: 225 nm (30,364) and 255 nm (10,471). Molar Conductivity: (10^−3^ M, CH_3_OH): 84.6 μS cm^−1^ mol^−1^.

### 2.4. Crystal Structure Determinations

The intensities for the X-ray determinations were collected at room temperature on a BRÜKER APEX II duo for HL3^FPh^, HL4^ClPh^, HL5^NO2Ph^ and HL6^Al^; and on a STOE IPDS 2T diffractometer at 200 K for the **Au1** and **Au2** complexes, both instruments equipped with Mo-Kα radiation (λ = 0.71073 Å). Standard procedures were applied for data reduction and absorption correction [35,36]. The structures were predicted by direct methods using SHELXS97 [37] or SHELXT [38] and refined by using SHELXL2016 [39], programs included in the OLEX2 program package [40]. The positions of the hydrogen atoms were calculated at idealized positions and treated with the “riding model” option of the SHELXL2016 program [39]. The details regarding structure refinement can be found in Table 1, with the exception of HL4^ClPh^, whose data were not adequate. Since **Au1** crystallized together with disordered solvent molecules (partially close to special positions), the refinement of its structure was undertaken with the removal of the disordered solvent molecules using the solvent mask option of OLEX2. Details are given in the Appendix A. The representations of molecular and crystalline structures were prepared with the programs Mercury 4.3.1. [41] and DIAMOND 4 [42].

### 2.5. Theoretical Calculations

#### 2.5.1. DFT

The geometry optimizations and harmonic frequencies calculations of all complexes were conducted using Density Functional Theory (DFT) [43]; the Becke functional, B3LYP, was applied at first [44]. For the silver complexes, the D3-BJ correction for dispersion effects was enclosed [45].

For the gold complexes, we used a methodology that includes relativistic calculations as stated by the Douglas–Kroll–Hess (DKH) scheme [46]. The recontracted basis set functions were applied following the scheme displayed in the Appendix A. All calculations were conducted in the polarizable continuous medium considering the dielectric constant of the solvent (ε = 36.6, acetonitrile) in the CPCM model [47]. Calculations were performed using ORCA software version 4.2.1 [48].

For the silver complexes, the basis set functions of Ahlrichs [49] def2-SVP were used for all atoms, except sulfur and silver, for which a def2-TZVP set was applied. All calculations were conducted in continuous medium considering the dielectric constant of the solvent (ε = 36.6, acetonitrile) in the CPCM model [47]. Calculations were performed using ORCA software version 5.0.2 [50].

#### 2.5.2. Molecular Docking

The docking simulations were performed using the optimized structures of the compounds in the enzyme active site with the GOLD (Genetic Optimization for Ligand Docking) suite version 5.5. All the water molecules and heteroatoms were removed from the enzyme and the flavin mononucleotide riboflavin monophosphate (FMN) prosthetic group was maintained in the catalytic site of enzyme structure. Only residues within 10.0 Å around His186 residue were used as a cavity site. Hermes was used for pre- and post-docking visualization, with the genetic algorithm (GA) method to run the calculations [51]. Full flexibility was allowed to the compounds. GA runs were conducted herein with a maximum of 100,000 GA operations were performed on a population size of 100 individuals. Diverse solutions were generated, ring corners were allowed to flip, conformations were explored and no constraint was applied to the protein or to the compounds. Redocking simulations of the FMN prosthetic group were executed in the enzyme active site with the available GOLD score functions and the best score (lowest *rmsd* to the X-ray conformation) was found with the GoldScore fitness function [51]. The enzyme used was based on the OLD Yellow protein from the protozoan *Trypanosoma cruzi* found in the database registered under the code PDB 4E2D [52].

### 2.6. Biological Studies

#### 2.6.1. Cultivation of the *Leishmania infantum* and *Leishmania braziliensis*

The *Leishmania infantum* and *Leishmania braziliensis* strains were maintained in Schneider medium supplemented with 20% of SFB (Schneider 20%), cultivated in 25 mm^3^ bottles and stored at 28 °C in a BOD oven [53]. For the assays, parasites at the 5th day stationary growth phase were used.

The promastigotes metacyclic form of *Leishmania infantum* and *Leishmania braziliensis* were first filtered with an insulin syringe to break up rosettes and, then, were isolated from the culture in the stationary phase by centrifugation of 400× *g* for 4 min at a temperature of 4 °C [53]. The supernatant was subjected to a new centrifugation, heavier, of 2000× *g* for 30 min, also at 4 °C. The supernatant was discarded and the cell pallet formed from this centrifugation was resuspended in 10 mL of 10% Schneider medium. The process ensured the removal of cellular debris and dead parasites, as well as possible crystals from the culture medium, so that the quality of the experiment was guaranteed. Then, the promastigotes were diluted to 1:10 in paraformaldehyde (PFA) for counting in a Neubauer Chamber, with the concentrations adjusted for each experiment.

An amount of 1 × 10^7^ parasites per well was used. The parasites were plated in 96-well plates. Then, the compounds were added in triplicate, starting at a concentration of 500 µM, with serial dilutions being performed up to a concentration of 3.9 µM [53]. The treatment was carried out for 24 h and, after adding 5 µL of resazurin, the fluorescence was read using a Enspire model spectrofluorometer (Perkin-Elmer, Waltham, MA, USA).

#### 2.6.2. Cytotoxicity Assays (Vero Cells Culture)

For the experiment, Vero cells (ATCC CRL-1587) were used in Roswell Parl Memorial Institute (RPMI) medium with 10% fetal bovine serum (FBS) (RPMI 10%) and 20 μg/mL of Gentamicin. They were cultivated in culture bottles of 25 mm^3^ and kept in an oven at 37 °C with 5% carbon dioxide (CO_2_). When picked or used on the day of the experiment, the cells were detached from the wall of the bottles with saline and 1X trypsin/EDTA. To find out how many cells there were in the bottle, it was diluted in Turkey liquid and then we proceeded with the counting in the Neubauer Chamber to calculate the desired concentrations [54].

The Vero cells were plated at 5 × 10^5^ cells per well, in 96-well plates, and kept in an incubator with 5% CO_2_ at 37 °C for 24 h. Then, the compounds were added in triplicate, starting at a concentration of 500 µM, with serial dilution being performed up to a concentration of 3.9 µM [54]. The treatment was carried out for 24 h, and, after adding 5 µL of resazurin, the fluorescence was read using a spectrofluorometer (EnSpire Perkin-Elmer, Waltham, MA, USA).

#### 2.6.3. Infectivity Test

For the infection of RAW macrophages adhered to the plates, we used parasites in the promastigote form of *Leishmania infantum*, in the stationary phase, in a ratio of 10:1 (promastigotes/macrophages) [55].

Macrophages were plated in the amount of 5 × 10^4^ cells per well, in 96-well plates, which were kept in a CO_2_ oven at 37 °C for 24 h. The parasites were plated at a concentration of 10:1. The plates were kept in a CO_2_ oven for 12 h until the parasites infected the cells. Then, we performed a cycle of 3 washes to remove the parasites that did not infect, with the treatment with the compounds being performed for 24 h. Plates were fixed with paraformol for further evaluation. Plaques were evaluated by counting parasites within 200 cells under the EVOS Cell Imaging System (Thermo Scientific, Waltham, MA, USA) microscope by staining with propidium iodide [55].

### 2.7. Enzymatic Studies

#### 2.7.1. Expression and Purification of LbOYE

The recombinant LbOYE protein was produced using the methodology previously described [56]. Briefly, the protein was expressed in strains of bacterial *Escherichia coli* BL21(DE3) through induction by IPTG using the pET28a:LbOYE vector. The enzyme was purified through a chromatography step of capture by nickel affinity, as the pET28a vector allows the expression of a polyhistidine fusion peptide, located at the N-terminal end of the protein. After affinity purification, the polyhistidine tail was excised by “overnight” incubation with thrombin, and the second purification step was performed using size exclusion chromatography. Purification was evaluated by SDS-PAGE and protein concentration was estimated by spectrophotometry [56].

#### 2.7.2. Suppression of Intrinsic Tryptophan Fluorescence

The emission spectra of the intrinsic fluorescence of the tryptophan were recorded using a Shimadzu F-4500 spectrofluorometer in the range of 310–420 nm, with opening of slits of excitation and emission at 5.0 nm and λ_ex_ = 295 nm. Interaction analyzes between LbOYE and ligands were performed at λ_max_ = 333 nm.

The compounds were solubilized in DMSO and then we made successive dilutions in a Tris-HCl 25 mmol L^−1^ buffer, pH 8.0, containing 100 mmol L^−1^ of NaCl. For all experiments, the final concentration of DMSO was kept constant and equal to 2.5%. Solutions with increasing concentrations of ligand and constant concentration of LbOYE (1 µmol L^−1^) were prepared, which were kept at 25 °C for a period of 1h before acquisition of fluorescence emission spectra.

The results were obtained from the fluorescence emission spectra of the intrinsic tryptophan probe in the presence of the compounds HL1^Ch^, HL3^FPh^, HL5^NO2Ph^, [Au(HL1^Ch^)_2_]Cl and [Ag(HL3^FPh^)_2_]NO_3_. The bonding parameters were determined by the Hill plot, obtaining the dissociation constant (*K_d_*) through the non-linear adjustment of the saturation curve to the experimental data. The double-log plot was used to determine the bonding constant and the number of bonding sites (n). It also allows the calculation of the bonding constant (*K_b_*) and *K_d_* from 1/*K_b_*. The Stern–Volmer plot at different temperatures demonstrates the static nature of the temper due to the increase in the temper with increasing temperature.

Equation (1)—Hill’s equation:(1)$$\frac{F_{0}-F}{F_{0}}=\frac{\left[L\right]}{\left[L\right]+K_{d}}$$

Reference: [21].

Equation (2)—Double-log:(2)$$log\left(\frac{F_{0}-F}{F_{0}}\right)=nlogK_{b}-nlog\left(\frac{1}{\left[L\right]-(F_{0}-F)[P_{0}]/F}\right)$$

Eeference: [57].

Equation (3)—Equation of Stern–Volmer:(3)$$\frac{F_{0}}{F}=1+K_{SV}[L]$$

Reference: [58].

In these equations, *F_0_* is fluorescence in the absence of any quencher, [*L*] is the ligand concentration and [*P_0_*] is the protein concentration.

The methodology used to determine the *K_d_* values presupposes the formation of the protein–ligand complex, which results in the suppression of fluorescence when compared to the protein in the absence of the suppressor.

The study of fluorescence quenching can be affected when the ligand has absorption bands that coincide with excitation and emission wavelengths of fluorescence [59]. This interference is called the internal filter effect, which can be minimized using the procedure described by Equation (4) [59,60]. *F_obs_* and *F_cor_* are the fluorescence intensities at 333 nm measured and corrected, respectively; *l* represents the optical step where the absorption and emission of light occurs and C_0_ is the concentration of the ligand.

For the correction of the internal filter effect, were used the molar absorptivity coefficient values in the excitation (295 nm) and emission (333 nm) lengths, [λ/ε (L mol^−1^ cm^−1^)]; HL1^Ch^ (*ε_ex_* = 373 and *ε_em_* = 117), HL3^FPh^ (*ε_ex_* = 1191 and ε_em_ = 57), HL5^NO2Ph^ (*ε_ex_* = 6162 and *ε_em_* = 9124), **Au1** (ε_ex_ = 7045 and *ε_em_* = 4008) and **Ag3** (*ε_ex_* = 9431 and *ε_em_* = 3281).

Equation (4)—Procedure to minimize the internal filter effect [59,60]:(4)$$F_{cor}=F_{obs}e^{-2.303\left(\varepsilon_{ex}+\varepsilon_{em}\right)lC_{0}}$$

#### 2.7.3. Enzyme Kinetics

The initial rates were obtained through the decay of the band at 340 nm referring to the maximum absorption of the NADPH (Nicotinamide Adenine Dinucleotide Phosphate) reducer, as previously reported [61]. Solutions containing 1 µmol L^−1^ of LbOYE, DMSO or 20 µmol L^−1^ of the tested compounds from a stock solution at a concentration of 5 mmol L^−1^ in DMSO were prepared in the presence and absence of the substrate *N*-Ethylmaleimide (NEM). These solutions were left for 30 min at room temperature. After 30 min, the NADPH reducer was added at a final concentration of 100 µmol L^−1^.

## 3. Results and Discussion

### 3.1. Synthesis and Characterization

The synthesis and spectroscopic characterization of HL^R^ ligands by FTIR and ^1^H NMR has already been reported and will not be the focus of the present work [22,32,33,34]. Herein, HL^R^ (R = F, Cl, NO_2_ and Alil) were further analyzed by other methods, including crystallographic determination. Furthermore, four new silver(I) complexes of general composition [Ag(HL^R^)_2_]NO_3_ (**Ag3**–**6**) and six new gold(I) complexes of composition [Au(HL^R^)_2_]Cl (**Au1**–**6**) were obtained, as presented in Figure 1. The reactions occurred in a 1:2 metal/ligand ratio in ethanol for the silver(I) complexes and in acetonitrile for the gold(I) complexes, under stirring at room temperature. The yields obtained were in the range of 54–76% for **Ag3**–**Ag6**, and around 80% for **Au1**–**Au6**. The colorless crystalline solids **Ag3**–**Ag6** are soluble only in DMSO, while the complexes **Au1**–**Au6** are partially soluble in a mixture of methanol or ethanol with dichloromethane (1:1) and fairly soluble in DMSO. The **Ag3**–**Ag6** and **Au1**–**Au6** complexes showed melting points in the range of 160–200 °C. The CHNS analyses were in accordance with the proposed structures of the **Ag3**–**Ag6** and **Au1**–**Au6** complexes. The poor solubility of **Ag3**–**Ag6** in lower viscosity solvents resulted in inaccurate conductivity values [62], whereas the molar conductivity values for the **Au1**–**Au3** and **Au6** complexes were in accordance with the formation of cationic compounds, except for the **Au4** complex, which differed from the other complexes due to the low solubility. The compounds were also studied by IR and ^1^H NMR; additional proof for the identity of the products is given by ESI^+^-MS for some representatives (see Appendix A for details).

### 3.2. Crystal Structures

The HL3^FPh^, HL4^ClPh^, HL5^NO2Ph^ and HL6^Al^ ligands were studied by single crystal X-ray diffraction. The molecular structures can be seen in Figure 2. Selected experimental bond lengths and angles for the ligands can be seen in Table 2, except those of HL4^ClPh^ as the data quality was not good enough to support a discussion about bond lengths and angles. HL3^FPh^, HL5^NO2Ph^ and HL6^Al^ were crystallized in the Triclinic crystal system and *P*ī space group. It was observed that the C(1)–N(2) and C(1)–N(3) bonds presented a shorter length (around 1.35 Å) when compared to C(9)–N(3) (around 1.43 Å), configuring an intermediate character between single and double bonds between the atoms N(2)−C(1)−N(3), while the S(2)−C(1) bond length had a double-bond character (around 1.67 Å).

The molecular structures for the gold complexes [Au(HL1^Ch^)_2_]Cl∙MeOH (**Au1**∙MeOH) and [Au(HL2^Ph^)_2_]Cl∙MeOH (**Au2**∙MeOH) were also determined using single crystal XRD. The representations of the **Au1** and **Au2** complexes can be seen in Figure 3. The ORTEP pictures may be observed in the Appendix A. The selected experimental bond lengths and angles are presented in Table 3. The complexes crystallized in the Monoclinic space group *I*2/*m*. Two thiosemicarbazide ligands coordinated via the sulfur atom in monodentate mode to the gold(I) metal ion. The geometry around the metal center was described as linear, however, a small distortion of the angle S(11)−Au−S(21) was detected 176.73(13)° for **Au1** and 175.16(9)° for **Au2** A likely explanation is that there is an interaction forcing the distortion. With regard to the bond lengths, it was noticed that the sulfur–carbon bond length S(11)−C(11) and S(21)−C(21) showed a small elongation in relation to the free ligands. Still, this bond length retained its double-bond character.

In the crystal lattice of the **Au1** and **Au2** complexes, hydrogen bonds involving N–H∙∙∙Cl, N–H∙∙∙O, O–H∙∙∙Cl and N–H∙∙∙O interactions can be found. For the **Au1** complex (Figure 4), the interactions involved the nitrogen atoms N(11) and N(12) as hydrogen donors to the chloride Cl(1) and oxygen O(1W) atoms, respectively, in addition to the nitrogen atom N(13) as a hydrogen donor to the oxygen atom O(10) from a neighboring unit, and the oxygen atom O(1W) as hydrogen donor to the nitrogen atom N(12) of a second neighboring unit. For the **Au2** complex (Appendix A), the interactions involved the nitrogen atom N(2) as a hydrogen donor to the oxygen atom O(1S), the nitrogen N(1) and oxygen O(1S) atoms with the chloride ion Cl(1), and the nitrogen N(3A) and N(3B) atoms as hydrogen donors to the oxygen atom O(1).

In addition to the hydrogen bonds, intermolecular interactions involving two Au(I) metal ions from different molecules were identified (see Figure 4). In the literature, many gold(I) complexes that present such type of interaction can be found [63,64]. According to the literature, the term “aurophilic interaction” was coined for this type of interaction, due to the high binding energy between Au∙∙∙Au ions, which resembles the energy found in intermolecular hydrogen bonds as well as to the low coordination number found for linear Au(I) complexes, which allows an opening in the coordination sphere of the metal ion for the approximation of another Au(I) metal ion [64]. Since there is no steric hindrance, the metal centers of different molecules were attracted by a distance equal to 3.8244(6) Å for **Au1** and 3.5268(6) Å for **Au2**, being in accordance with the expected distance for a weak Au∙∙∙Au intermolecular interaction (between 3.50–3.80 Å) [64].

### 3.3. Photophysical Studies

The electronic spectra of the **Ag3**–**Ag6** and **Au1**–**A6** complexes were obtained from freshly prepared solutions in acetonitrile or methanol (Appendix A). The electronic spectra data can be found in Appendix A and in the experimental part. Two absorption bands with maximum of absorption around 230 and 260 nm were observed for the **Ag3**, **Ag4**, **Ag6**, **Au1**–**Au4** and **Au6** complexes, while the **Ag5** and **Au5** complexes presented three absorption bands around 220, 270 and 350 nm. There were no significant changes in the bands observed in relation to HL1-6 free ligands; however, the molar absorptivities registered were different after complexation. The first band observed in the spectra can be attributed to π→π* intraligand transitions, while the second band, according to theoretical calculations obtained for similar silver(I) complexes [22], is characteristic of charge transfer transitions MLTC (metal-to-ligand charge transfer), due to the electronic configurations of Ag(I) and Au(I).

The second part of the electronic studies was to obtain the fluorescence emission spectra of the **Au1** and **Au2** complexes (Appendix A). The measurement at room temperature of the complexes was performed in methanol solutions with and without the Ar atmosphere; however, no apparent change was verified for the two complexes in the different conditions, indicating that the **Au1** and **Au2** complexes do not show luminescence at room temperature. Then, the measurement of emission spectra at low temperature (Appendix A) was carried out, which showed bands with emission maxima at a relatively high intensity around 400 nm. As the non-radiative decay of the excited state was strongly suppressed, the luminescence of lower excited states could be detected. The characteristic of the extended bands observed is typical of MLTC emissions [65,66].

### 3.4. DFT Calculations

Aiming to better understand the properties of the compounds, theoretical calculations (DFT) were performed for the monomers of the Ag^I^ and Au^I^ complexes. The following parameters were verified: the υ(C=S) band region, bond lengths and angles, the energy gap and composition of the orbitals (HOMO-LUMO) participating in the transitions that occurred from the state fundamental for the excited state of the complexes, as well as the AIM (Atoms in Molecules) data obtained by the optimized structures of the complexes.

Two different basis sets were used since gold and silver are classic examples of the importance of relativistic effects, as the difference between them is mainly due to these effects. Despite the use of an effective core potential to recover some of the relativistic effects for an element like gold, the use of a Hamiltonian with the inclusion of relativistic effects results in a better description of its properties [67,68,69,70]. In addition, in the case of the silver complexes the basis set def2-TZVP was used only for Ag and S, while def2-SVP was used for remaining atoms. The description of the metal with basis set functions of a triple-zeta character is important for a better description of its coordination chemistry, with several bonds being made. On the other hand, there are studies in the literature showing that the use of double-zeta character basis functions is sufficient for a good description of the ligands, in addition to resulting in a lower computational cost [71].

After optimizing the structures, the vibrational spectra were obtained. It was analyzed whether the assignment of the υ(C=S) band obtained by the experimental spectra of the Au^I^ complexes was in accord with that expected by the theorical calculations. The attributions can be seen in Appendix A. The frequencies presented in the table were not scaled in relation to the experimental spectra. There is no “pure” vibrational mode only for the υ(C=S) stretch, but conjugated modes as assigned in the table. According to theoretical data and experimental spectra, it is suggested that the stretching was within the expected region (730–800 cm^−1^), but with a variation between each complex. Then, from the optimized structures of the Ag^I^ and Au^I^ complexes (Appendix A) the theoretical bond lengths and angles were obtained (Appendix A). For the **Au1** and **Au2** complexes, determined experimentally by X-ray diffraction and discussed earlier, the Au-S bond lengths and S-Au-S angle were consistent with the calculated values for both complexes.

The HOMO-LUMO orbitals of the Au^I^ and Ag^I^ complexes (Appendix A, respectively), as well as the energy gap between the orbitals (Table 4) were also calculated. It was observed that in the Au^I^ and Ag^I^ complexes with the HL1^Ch^ and HL6^Al^ ligands, the transition occurs from the HOMO orbital, with predominant composition in the C=S and N-H bonds, to the LUMO orbital, with the majority composition of the p-toluene group. On the other hand, the complexes derived from the HL2^Ph^, HL3^FPh^ and HL4^ClPh^ ligands, the peripheral groups compose the HOMO orbital, while the LUMO orbital remains primarily in the p-toluene group. In the case of the complexes with the HL5^NO2Ph^ ligand, the nitrophenyl group participates in both the HOMO and LUMO orbitals.

Regarding the energy gaps (between HOMO and LUMO) of the complexes, it was observed that the energies follow the same order of explanation of the orbitals. In descending order, the highest gap values were found for Ag^I^ and Au^I^ complexes with HL1^Ch^ and HL6^Al^ ligands. Then, the values of the Ag^I^ and Au^I^ complexes with the HL2Ph, HL3^FPh^ and HL4^ClPh^ ligands were found, and finally, the lowest values were observed for the Ag^I^ (**Ag5**) and Au^I^ (**Au5**) complexes with the H**L5^NO2Ph^** ligand. The different energy gaps can be attributed to the different substituent groups of the ligands in the complexes, and therefore, the lowest gap energy value was found with the complexes that have the nitrate substituent group in the ligand, since it comes to a group more puller of electron density.

Finally, the AIM (Atoms in Molecules) data of the Au^I^ complexes (Appendix A) were also obtained that comprise: the Bader charge (q), the delocalization index (DI), the electronic density at the relevant binding critical point (ρ), and the Laplacian of the density at this point (∇^2^ρ). The DI values are very indicative to say that each bond really has the characteristic of a single covalent bond.

The electron densities vary fairly between compounds due to the nature of each ligand, but the influence is relatively small on atoms on the coordination sphere of gold(I). With regard to the Bader charges, they evaluate the charges of metal ions and coordinated sulfur atoms. As expected, and according to the change in substituent on the ligands in each Au^I^ complex, the charges show a small change from one complex to the other. The most positive charges were found for the **Au5** complex (q = Au^I^: +0.1255, S3: +0.1277 and S4: +0.1269) with the HL5^NO2Ph^ ligand, while the least positive charges were found for the complexes **Au1** (q = Au^I^: +0.1028, S3: +0.0588 and S4: +0.0589) and **Au6** (q = Au^I^: +0.1078, S3: +0.0610 and S4: +0.0606) with the HL1^Ch^ and HL6^Al^ ligands, respectively. It is noticed that the sulfur atoms suffer a much more significant decrease in the charge than the charges of the metal ions. In addition, the Bader charges follow an opposite trend in relation to the energy gaps of the orbitals. For the **Au5** complex, a smaller energy gap and a more positive Bader charge was determined, while for the **Au1** and **Au6** complexes, a larger energy gap and a less positive Bader charge were calculated.

### 3.5. Biological Studies

#### 3.5.1. Cytotoxicity on Vero Cells

Based on the interest in developing new drugs for Leishmaniasis, biological studies were initiated by carrying out in vitro tests to evaluate the cell viability (cytotoxicity) of free ligands (H**L1-6**) and **Ag3**–**Ag4** and **Au1**–**Au6** complexes against kidney cells of the African green monkey (*Vero*). In addition to these compounds, the [Ag(HL1^Ch^)_2_]NO_3_ (**Ag1**) and [Ag(HL2^Ph^)_2_]NO_3_ (**Ag2**) complexes, obtained in a previous work, were included in the test [22]. Through this test, it was possible to determine the CC_50_ values (inhibitory concentration for the viability of 50% of the cells). The higher the inhibitory concentration the greater the viability of the compound against the Vero cells. The CC_50_ values can be found in Table 5 and the graphs of viability in function of the concentrations are found in Appendix A. Analyzing the graphs obtained for all compounds, it was observed that the free ligands present good cytotoxicity indexes (between 200–300 μM), with the exception of the HL4^ClPh^ and HL5^NO2Ph^ ligands. Among the complexes tested, four silver(I) complexes, **Ag3**–**Ag6**, showed CC_50_ values close to or above 500 μM, while only one gold complex, [Au(HL2^Ph^)_2_]Cl (**Au2**), showed good cytotoxicity (295.6 µM). The **Au2** complex was less toxic than its free ligand HL2^Ph^, while the **Ag3**–**Ag6** complexes showed significantly better toxicities than their respective free ligands. For the other complexes, it was observed in general that the complexes presented a higher cytotoxicity than the free ligands.

#### 3.5.2. Tests against *Leishmania infantum* and *Leishmania braziliensis* Parasites

The leishmanicidal activity of the compounds was evaluated against the promastigote form of *L. infantum* and *L. braziliensis*. Glucantime was used as reference drug. Appendix A show the graphs referring to the results of the leishmanicidal activity versus tested concentrations of the compounds. The IC_50_ values can be seen in Table 5.

For the *L. braziliensis*, it was observed that the HL1^Ch^, HL4^ClPh^ and HL5^NO2Ph^ showed better activity than the reference drug (176.3–191.7 μM), and the HL5^NO2Ph^ presented an excellent result with an IC_50_ value equal to 19.02 µM. Three silver(I) complexes **Ag1**, **Ag4** and **Ag5** were more active for this strain than reference drug, with **Ag1** the most active among them (IC_50_ equal to 50.15 μM). Following the same trend, the gold(I) complex with the HL1^Ch^ ligand, **Au1**, showed the best activity (IC_50_ = 77.38 μM).

For the *L. infantum* strain, it was possible to verify that the free ligands HL1^Ch^ (11 μM) and H**L2^Ph^** (48.39 μM) presented the best activities when compared to the activity of Glucantime (113.2–130.2 μM). All Ag(I) complexes, with the exception of **Ag2**, showed a good activity, highlighting the **Ag1** (57.42 μM) and **Ag4** (72.17 μM) complexes. Regarding the Au(I) complexes, **Au2**, **Au4** and **Au6**, with values of IC_50_ of 41.69, 38.18 and 62.54 µM, respectively, showed the most promising results. A similar activity was found for other antileishmanial compounds like primaquine (26.5 ± 1.2 μM), miltefosine (14.4 ± 1.1 μM), and [AuCl(PEt_3_)] (16.59 ± 1.03 μM) and auranofin (9.68 ± 1.02 μM), though conditions and methods vary [29,72].

The obtained data indicate that the peripheral groups of the ligands can influence the biological activity of the complexes, presenting different results from each other. Furthermore, the biological performance of the compounds is different between the two tested strains. A similar behavior was found for gold(I) complexes containing *N*-heterocyclic carbenes with *L. amazonensis* and *L. braziliensis* [73]. It is well-known that the properties of bioactive organic compounds are influenced by metal coordination. In most cases, an increased activity of the complexes is observed, which is assumed to be a modulation of pharmacokinetic parameters caused by a metal-assisted transport, while the complex dissociation inside the cell releases the ligands as the biologically active species [28].

In order to have a better toxicity response against the parasites, the IC_50_ should be as low as possible, while to have a better cytotoxicity response, the CC_50_ should be as high as possible. The ratio between the CC_50_ and IC_50_ values expresses the selectivity index (SI) of the compounds and indicates which of these are promising for the development of future drugs. Table 5 presents the IC_50_, CC_50_ and SI values of the compounds.

**Table 5 pharmaceutics-16-00452-t005:** Values of CC_50_ (µM) on Vero cells, IC_50_ (µM) against the promastigote form of the *L. braziliensis* and *L. infantum*, and selectivity indexes (SI) obtained for the compounds.

Compound	CC_50_ (µM)	IC_50_ (µM) *L. braziliensis*	IC_50_ (µM) *L. infantum*	SI *L. infantum*	SI *L. braziliensis*
HL1^Ch^	293.0	128.9 ± 0.051	11 ± 0.306	26.64	2.27
HL2^Ph^	268.0	273.5 ± 0.018	48.39 ± 0.189	5.54	0.98
HL3^FPh^	291.8	204.5 ± 0.018	201.2 ± 0.052	1.450	1.43
HL4^ClPh^	16.80	133.8 ± 0.036	108.4 ± 0.104	0.16	0.13
HL5^NO2Ph^	26.62	19.02 ± 0.012	286.9 ± 0.075	0.09	1.40
HL6^Al^	187.9	313.5 ± 0.063	812.8 ± 0.515	0.23	0.60
**Ag1**	7.400	50.15 ± 0.020	57.42 ± 0.028	0.129	0.15
**Ag2**	92.14	183.7 ± 0.027	280.5 ± 0.167	0.33	0.50
**Ag3**	>500	233.9 ± 0.031	92.83 ± 0.072	>5.39	>2.14
**Ag4**	>500	84.62 ± 0.116	72.17 ± 0.036	>6.93	>5.91
**Ag5**	493.3	90.73 ± 0.236	78.79 ± 0.091	6.26	5.44
**Ag6**	>500	305.2 ± 0.032	78.59 ± 0.605	>6.36	>1.64
**Au1**	19.94	77.38 ± 0.018	105.1 ± 3.139	0.19	0.26
**Au2**	295.6	143.2 ± 0.085	41.69 ± 0.210	7.09	2.06
**Au3**	28.77	194.9 ± 0.063	173.9 ± 0.087	0.17	0.15
**Au4**	31.98	277.6 ± 0.163	38.18 ± 0.158	0.84	0.12
**Au5**	58.59	284.0 ± 0.259	295.5 ± 0.259	0.198	0.21
**Au6**	77.52	5554 ± 1.371	62.54 ± 0.151	1.24	0.01
Glucantime	1634 [74]	176.3–191.7	113.2–130.2	14.43–12.55	9.26–8.52

Five compounds presented good SI values (HL1^Ch^ = 26.6, HL2^Ph^ = 5.5, **Ag3** ≥ 5.39, **Ag4** ≥ 6.93, **Ag5** ≥ 6.36 and **Au2** = 7.09) for the *L. infantum* promastigotes. Compared to the SI of Glucantime, the most promising result was for the HL1^Ch^. The Ag(I) complexes were in general more selective than the Au(I) complexes, with **Au2** being the only one with a significant SI. In relation to the *L. braziliensis* promastigotes, only two compounds showed good SI values (**Ag4** = >5.91 and **Ag5** = 5.44). This result is comparable to that of amphotericin B, whose SI is 6.8 considering the activity against the same leishmania species and the cytotoxicity on macrophages [73]. On the other hand, the ligands did not display a good selectivity for *L. braziliensis*. Following the same trend of the tests against *L. infantum*, the silver(I) complexes were the most selective for this parasite. Generally, silver complexes display lower selectivity towards cancer cells compared to Au(I) complexes, as they present an increased lability resulting in a faster release of Ag^+^ ions; however, any assumption at this stage would be mere speculation [75,76].

In view of the data obtained, the six most promising compounds (HL1^Ch^, HL2^Ph^, **Ag3**, **Ag4**, **Ag6** and **Au2**) were selected for carrying out the infection rate test against the *L. infantum* parasite. For this test, the *L. infantum* parasites and the compound to be evaluated in concentrations based on the IC_50_ values (the obtained value, one value below and one above) were plated with the Vero cells. This test was performed at four stages. The first step determined the percentage of the amount in 200 infected macrophages per parasite, the second determined the percentage of parasites that infects 100 cells/macrophages, the third determined the average percentage of parasites that infects 200 cells/macrophages and, finally, the fourth step indicated the infection rate, that is, the result of the amount of Leishmania per macrophage (obtained from the third step) multiplied by the percentage of infected macrophages. Figure 5 presents the graphs of the infection rates, while the other graphs can be found in Appendix A. From these graphs, it was observed that all compounds inhibited the parasite infection at the three concentrations (IC_50_<, IC_50_ and >IC_50_). The compound HL1^Ch^, in addition to inhibiting the infection, considerably lowered the infection rate of the parasite even at a concentration below its IC_50_. For HL2^Ph^, the infection index was not relevant for a concentration below its IC_50_. Regarding the complexes, **Ag4** did not show a promising infection rate, while the **Ag3**, **Ag6** and **Au2** complexes were able to inhibit the parasite infection inside the cells even at lower concentrations than that determined by their IC_50_ values.

### 3.6. Enzymatic Studies

After the synthesis and structural characterization of the compounds, studies of interaction of the compounds and the recombinant LbOYE were carried out by fluorescence measurements. Initially, the solubility in buffer at the appropriate concentrations for carrying out the fluorescence experiments was evaluated; therefore, only the compounds that passed this stage were selected, them being HL1^Ch^, HL3^FPh^, HL5^NO2Ph^, **Ag3** and **Au1**.

The fluorescence emission was obtained at increasing concentrations of HL1^Ch^, HL3^FPh^ and HL5^NO2Ph^, which caused a decrease in the fluorescence emission intensity for all studied ligands. The quenching caused by the presence of the compound H**L5^NO2Ph^** (Appendix A) was more pronounced than when compared to HL1^Ch^ and HL3^FPh^ (Appendix A). This difference indicates a lower affinity of the LbOYE for the HL1^Ch^ and HL3^FPh^.In this way, it was possible to adjust the Hill equation for the determination of K_d_ only for the ligand HL5^NO2Ph^. The Stern–Volmer relationship states that the kinetic study of the photophysical deactivation of the fluorophore can be performed. The nature of this deactivation can be static or dynamic [58,77]. The static deactivation occurs when the interaction results in the formation of the protein–compound complex, which is reflected in the decrease in fluorescence intensities. The fluorescence quenching can also result from deactivation by collision between molecules, which is called dynamic quenching, which is also favored by the increase in temperature [58]. In addition, the latter can occur together with static suppression when in the presence of high concentrations of the compound [58].

Figure 6 shows the results of the interaction between **Ag3** and the LbOYE protein. The **Ag3** complex showed interaction with K_d_ in the µM order and the Hill curve reaches its saturation which causes an increase in molecules in solution that do not interact with the protein, leading to a deviation from linearity in the Stern–Volmer graph and an increase in dynamic processes. **Au1** shows interaction with the LbOYE enzyme as exposed in Appendix A. The temperature variation showed a slight increase in the suppression for higher concentrations where the sum of static and dynamic effects can be observed.

The data obtained by the mathematical treatments applied to the experimental data (Table 6) demonstrate that, among the organic compounds, HL5^NO2Ph^ presented the highest affinity. Among the complexes, **Ag3** showed the highest affinity, although its ligand HL3^FPh^ showed low affinity for the LbOYE protein. The same effect was observed for **Au1**, which showed an increase in the affinity in relation to the HL1^Ch^ ligand.

After studying the interaction with the enzyme, the influence of the presence of the **Au1** and **Ag3** complexes on the LbOYE enzymatic activity was determined using two substrates, O_2_ and the *N*-Ethymaleimide (NEM). As a control, their HL1^Ch^ and HL3^FPh^ ligands were also tested.

The inhibition effect can be identified by the decrease in the initial velocity V_0_, which is obtained by the decay of the NADPH reducing band at 340 nm. Figure 7 evidences that the **Ag3** was capable of inhibiting the reaction of the enzyme both with the NEM and for O_2_. The **Au1** also demonstrates capacity of inhibition, although less than the **Ag3**. The respective free ligands did not influence the reaction velocity with the NADPH.

### 3.7. Molecular Docking Studies

With the purpose of finding new bioactive agents and providing a possible mechanism of action for the compounds described in this work, a Docking analysis with the OLD Yellow enzyme of the *Leishmania braziliensis* was performed. The enzyme used was based on the OLD Yellow protein from the protozoan *Trypanosoma cruzi* as they have the same structure, and can be found in the database registered under code PDB 4E2D [52]. The Old Yellow enzyme has a prosthetic group called Flavin Mononucleotide, known by its acronym FMN. In order to exert the activity of the enzyme, it is necessary for the FMN to pass from its oxidized form to the reduced form. For this, FMN uses as a cofactor NADPH, which reduces it to FMNH_2_ through electron transfer [21]. Our assumption is that the metal complex can target this enzyme by preventing this process to happen, capturing electrons from FMN or NADPH.

The purpose of the molecular docking performed was to identify the most favorable conformation of each compound (ligand) in the active site of the enzyme as well as the residues participating in the interactions, and thus determine the receptor–compound bond energies (score) [78,79]. In addition to comparing the activity between the silver(I) and gold(I) ions of the analogous complexes, the alteration of the peripheral groups of the ligands is also a factor that can influence the interaction between the molecule and the enzyme. In this way, representative figures of each molecule at the site of the LbOYE were obtained (Figure 8 and Appendix A).

It was observed that some compounds are docked close to amino acid residues responsible for the enzyme–substrate interaction. All free ligands, the silver(I) complexes **Ag1** and **Ag2**, and the gold complex **Au3** were found next to the Tyr352 residue. The **Ag1** complex is also close to residues Phe279 and Asp134, and the **Ag2** complex close to the residue Thr32. In addition, it was observed that **Ag4** and **Au2** interact with the His135 and with Ser244 residues, respectively, while the other complexes did not interact with any residue. Thus, it was shown that the interaction does not depend only on the metal, but on the structure as a whole. Furthermore, the complexes fit into the space within the active site of the enzyme, almost parallel to the FMN structure. This indicates that they are in a suitable position to block the process of electron transfer from NADPH to FMN. It was verified that the HL1^Ch^ and HL2^Ph^, **Ag4** and **Au4** were the closest to the FMN.

The energy corresponding to the interaction between the enzyme and the substrate is called bond energy [80]. From the most energetically favorable conformations within the site, the scoring values of the compounds (Table 7) were obtained. A lower bond energy (scoring) means a better interaction of the molecule with the enzyme. Thus, the compounds that energetically showed better interactions were HL6^Al^ and its respective silver and gold complexes, **Ag6** and **Au6**. Furthermore, the highest energy gap values (Table 4) were also found for the **Ag6** and **Au6** complexes. The energy gap obtained by DFT calculations determines the ability that a compound to donate or receive electrons [81]. If the difference is large, it is in principle a more stable complex in relation to the oxidation process. Thus, it is suggestive that these complexes are good at disrupting the electron transfer from the FMN group to the NADPH. However, there are cases in which the compound can only inhibit the protein by blocking the active site and still not participate in the electron transfer process with the FMN prosthetic group [80].

## 4. Conclusions

Metal complexes of composition [M(HL^R^)_2_]X (M = Ag or Au; X = NO_3_^−^ or Cl^−^) were successfully synthesized and characterized by different techniques, including the crystallographic determination of the **Au1** and **Au2** complexes and DFT calculations, which allowed us to not only confirm the structures of the molecules but also to understand their electronic properties.

The compounds were evaluated in vitro on Vero cells (Vero) in order to determine their cytotoxicity and against two Leishmaniasis strains, *Leishmania infantum* and *Leishmania braziliensis*. The activity tests showed a different behavior of the compounds against the two strains showing a dependence on both the metal center and peripheral groups of the ligands. The cytotoxicity tests showed that, in general, the gold(I) complexes are more toxic than the free ligands and silver(I) complexes. Overall, some compounds displayed good selectivity indexes for both strains. The infection rate of *Leishmania infantum* determined for the **Ag3**, **Ag6** and **Au2** complexes showed that all of them are able to inhibit the parasite infection even at lower concentrations than that of their IC_50_ values.

The enzymatic studies showed that the compounds have affinity with the target Old Yellow Enzyme from *Leishmania braziliensis*. Among the evaluated organic compounds, HL5^NO2Ph^ showed the best interaction with the protein. The complexes [Ag(HL3^FPh^)_2_]NO_3_ (**Ag3**) and [Au(HL1^Ch^)_2_]Cl (**Au1**) showed a relatively higher affinity for the LbOYE protein when compared to free ligands, with the Ag^I^ complex having the best interaction. The enzymatic tests showed that the complex **Ag3** was also the one that best inhibited the reaction responsible for the enzyme activity. Furthermore, the docking analysis showed that changing the peripheral groups of the ligands affects the interactions at the Old Yellow enzyme active site of *L. braziliensis*. The compound HL6^Al^ and its complexes of Ag^I^ and Au^I^, **Ag6** and **Au6**, presented the lowest score values and, consequently, the best interactions with the target enzyme, consistent with the highest energy gap values obtained for them, which indicates that these complexes are more suitable for preventing the electron transfer from the NADPH to FMN group. However, because they are not soluble in the buffer used for the experiment, it was not possible to carry out the test experimentally. Altogether, with the present data it is possible to say that the class of compounds studied here may be considered as a platform for the development of a new agent for *Leishmaniasis* treatment, particularly considering the *Leishmania infantum* species.

## Data Availability

The data can be requested from authors.

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
