# Peer review of "Gold(I) and Silver(I) Complexes Containing Hybrid Sulfonamide/Thiourea Ligands as Potential Leishmanicidal Agents"

_pharmaceutics, 2024, doi:10.3390/pharmaceutics16040452_

Round 1

Reviewer 1 Report

Comments and Suggestions for Authors

The article titled “Gold(I) and silver(I) complexes containing hybrid sulfonamide/thiourea ligands as potential leishmanicidal agents” by Alice P. Borges,  Malu M. S. Obata,  Silvia H. Libardi,  Rafael O. Trevisan, Victor M. Deflon, Ulrich Abram, Francis B. Ferreira, Luiz Antônio S. Costa, Antonio O. T. Patrocínio, Marcos V. da Silva,  Júlio C. Borges, and Pedro I. S. Maia

I recommend publishing the paper in the Pharmaceutics journal after some additions.

The author synthesized several Ag and Au complexes as potential leishmaniasis agents and has described their synthesis and spectroscopic data. The biological tests have demonstrated that some synthesized compounds present remarkable leishmanicidal activity, even higher than that of the Glucantime standard drug with different selectivity for the species of Leishmania.

The part of the abstract about the HR MS measure is not true, as in supplementary material the authors give only MS spectra with ESI method.

In the part “2.3 Preparation of the compounds” I recommend including the information about MS measure concerning only the molecular ion of new derivatives of Ag complexes like for Au complexes.

Could you include more information on the explanation of the difference of the activity between free ligands and their complexes or silver and gold complexes?

Could you also interpret why the gold complexes are more toxic?  

After taking into account my suggestions and corrections, the paper is suitable to be published in the journal.

Reviewer 2 Report

Comments and Suggestions for Authors

This manuscript presents the synthesis and biological activity against leishmania of a series of metal complexes with sulfonamide/thiourea ligands.

The topic fits well with the scope of this journal.

The synthesis is simple and well described and the structural studies included melting points, RMN, IR, MS, elemental analyses, IR, UV, and x-ray.

The supplementary materials file containing the experimental data is of good quality.

The compounds used were also investigated using computational methods and docking studies.

The biological studies are also comprehensive and very detailed.

The discussion is clear, and the conclusions are supported by the data presented.

Overall, this is a good manuscript that can be published as is, since no obvious mistakes were found.

Reviewer 3 Report

Comments and Suggestions for Authors

The manuscript titled "Gold(I) and silver(I) complexes containing hybrid sulfonamide/thiourea ligands as potential leishmanicidal agents" by Maia and collaborators is a well-written and organized document that could be of interest to the scientific community interested in the design of new molecules potentially useful in globally neglected diseases such as Leishmaniasis. Nonetheless, some problems should be resolved before publication in this prestigious journal, for example:

The four ligands used have already been previously published, as reported by Scifinder. At least indicate it in the text and cite them in the manuscript.

·      Schenone, S.; Bruno, O.; Ranise, A.; Bondavalli, F.; Filippelli, W.; Falcone, G.; Giordano, L.; Vitelli, M. R. 3-Arylsulphonyl-5-arylamino-1,3,4-thiadiazol-2(3H)ones as anti-inflammatory and analgesic agents. Bioorganic & Medicinal Chemistry (2001), 9(8), 2149-2153.

·      Nalavde, Yogesh Madhukar; Joshi, Vidya. Synthesis of new substituted sulfonylhydrazinecarboxamides and sulfonylhydrazinecarbothioamides having antifungal and antibacterial activities. Indian Journal of Chemistry, Section B: Organic Chemistry Including Medicinal Chemistry (2000), 39B(1), 76-79.

·      Upadhyaya, J. S.; Upadhyaya, S. K. Thin-layer chromatographic separation of the potential antituberculous agents N-aryl-N'-pmethylbenzenesulphonyl Thiosemicarbazides. Journal of Chromatography (1978), 152(2), 595-596.

·      Asis, Silvia E.; Bruno, Ana M.; Gaozza, Carlos H. Synthesis and antineoplastic activity of 1-(4-toluensulfonyl)-4-arylsemicarbazides. Acta Farmaceutica Bonaerense (1997), 16(4), 209-214.

·      Sharma, Mahendra Pal; Upadhyaya, J. S.; Srivastava, S. P.; Swami, R. K. Potential antineoplastic agents. V: Synthetic and pharmacological studies on some N-aryl-N-(substituted) thiosemicarbazides. Indian Journal of Pharmaceutical Sciences (1981), 43(5), 177-179.

·      Munshi, A. A.; Shah, N. M.; Trivedi, J. P. Synthesis of arylsulfonyl hydrazones and 1-arylsulfonyl-4-substituted thiosemicarbazides. Indian Journal of Chemistry (1963), 1(7), 311-313.

·      Peretyazhko, M. Z.; Pel'kis, P. S. Synthesis of 2-arylsulfonylhydrazones. 3-Aryl-1,3-thiazolidine-2,4-diones. Ukrainskii Khimicheskii Zhurnal (Russian Edition) (1969), 35(5), 532-535.

·      Silberg, Al.; Proinov, I. Additions to senevols. III. The addition of some arylsulfonylhydrazides to isothiocyanates and the preparation of some arylsulfonylhydrazones. Acad. rep. populare Romîne, Filiala Cluj, Studii cercetari chim. (1959), 10, 329-334.

Additionally, some coordination complexes with metals of the first transition series were previously reported. These references could improve the introduction.

·      El-Asmy, Ahmed A.; Babaqi, Abdallah S.; Al-Hubaishi, Ahmed A. Ligational, corrosion inhibition and antimicrobial properties of 4-phenyl-1-benzenesulfonyl-3-thiosemicarbazide. Transition Metal Chemistry (Dordrecht, Netherlands) (1987), 12(5), 428-31

·      Doering, M.; Uhlig, E.; Undeutsch, B.; Gloe, K.; Muehl, P. Sulfonamide substituted thiono compounds as ligands in copper(II) chelates and as extractants for the late 3d-element ions. Zeitschrift fuer Anorganische und Allgemeine Chemie (1988), 567, 153-60

·      Doering, M.; Uhlig, E.; Nefedov, V. I.; Salin, I. V. Complex formation by sulfonamide substituted thionoligands. Zeitschrift fuer Anorganische und Allgemeine Chemie (1988), 563, 105-15

It is unusual to publish a research article with twelve authors from seven different institutions. What was the role of each author?

The words thiourea, oximes, carbazates, thiosemicarbazides and sulphonamides, as well as hybrid compounds, are not part of the keywords and are the ligands that support this manuscript.

Review the language and writing of the manuscript sentence by sentence and paragraph by paragraph. This would improve the quality of the document.

In the introduction check lines 94 and 95... "for the HLR derivatives brought us the interest in seeking to verify the influence of alterations both" …

In the abstract. the authors state that they characterized the compounds by FTIR, NMR, UV-Vis, emission, HRMS, EA, SC-XRD. However, there is no information about some equipment in the experimental part.

In table 1 indicate the size, shape and colour of the crystal. Include the deposit number in the Cambridge structural database.

In line 127 of methodology, the four ligands are referred to as new compounds. The same in line 472 of results. However, they have already been reported previously.

It would be extremely important to report the 13C NMR of the four ligands at least, this would improve the quality of the publication. On the other hand, 2D NMR could help to unequivocally characterize each carbon and each hydrogen of the structures.

The authors indicated that HRMS was used in the identification of all compounds. However, there is no evidence of this characterization technique in either the main text or the supplementary information for the ligands and their silver derivatives. It was only used in gold derivatives. How do you explain it?

Include CheckCIF Platon reports (supplementary information) to verify the quality of crystallographic data and the handling of type A and B errors.

The authors mention "the low solubility of the Ag3-Ag6 and Au4 complexes resulted in inaccurate conductivity values". Support it with bibliographic evidence

Homogenize the quality of figures 2, 3, and 4 using the same colour pattern for the atoms.

Improve the discussion of the results obtained in antiparasitic and cytotoxicity assays. There is no comparative analysis with other metallodrugs used against Leishmania.

Verify that all references comply with the specifications of the author guide.

Is it possible to obtain figure 8 with a white background?

Comments on the Quality of English Language

Minor editing of English language required

Reviewer 4 Report

Comments and Suggestions for Authors

The research focuses on the synthesis of gold(I) and silver(I) complexes, utilizing hybrid sulfonamide/thiourea ligands with the purpose of identifying potent leishmanicidal agents, given the inadequacies of current treatments. The study points on developing complexes as novel remedies for leishmaniasis that consists in a severe disease in tropical and subtropical areas. Ligands were synthesized  to form  a plethora of complexes coordinating gold (six cationic species) or silver centres (more six cationic species), all characterized by using different spectroscopic techniques. (IR,NMR,molar conductiviy, elemental analysis, ESI+Ms and SC-XRD). Assessment of cell cytotoxicity and leishmanicidal activity has also been conducted. The findings revealed that certain compounds exhibiting notable leishmanicidal effects, overcoming standard drug effects. Additionally, the interaction between complexes and the specific enzyme LbOYE of Leishmania braziliensis was examined. The research indicates the potential of these complexes as therapeutic agents against leishmaniasis, thereby contributing to the quest for innovative treatments against this neglected tropical disease.

line 61 the Authors must revise the sentence which has not significance

line 62 after trials , dot is necessary

Most important for the paper to be published are the  comments about DFT calculations: 

It is not clear why different basis sets have been used for gold and silver complexes, since the def2-TZVP basis set, used for Ag, is defined also for Au.  In addition, the use of bases of different quality for silver complexes (def2-TZVP for Ag and S, def2-SVP for remaining atoms) is known to be hazardous and must be at least justified  or homologated to the same more accurate calculation basis.
